# Acceptability and perceived barriers to reactive focal mass drug administration in the context of a malaria elimination program in Magude district, Southern Mozambique: A qualitative study

Carlos Eduardo Cuinhane[1,2]*, Beatriz Galatas[2,3], Julia Montaña Lopez[2], Helder Djive[2], Hoticha Nhantumbo[2], Ilda Murato[2], Francisco Saúte[2], Pedro Aide[2,4], Khátia Munguambe[3,5], Neusa Torres[2]

1 Department of Sociology, Faculty of Arts and Social Sciences, Eduardo Mondlane University, Maputo, Mozambique, 2 Centro de Investigação em Saúde de Manhiça, Maputo, Mozambique, 3 ISGlobal, Hospital Clinic–Universitat de Barcelona, Barcelona, Spain, 4 National Institute of Health, Ministry of Health, Maputo, Mozambique, 5 Faculty of Medicine, Eduardo Mondlane University, Maputo, Mozambique

* c.cuinhane@hotmail.com

**Data Availability Statement:** The data of this study were collected under individual-level informed

## Abstract

This study analysed acceptability and perceived barriers to reactive focal mass drug administration (rfMDA) among community members exposed to community engagement campaigns and malaria elimination interventions in Magude district, following mass drug administration (MDA) in the same district. The study used a formative qualitative study design, consisting of 56 semi-structured interviews with community members, including community leaders, household heads, women of reproductive age, members of the community and adolescents, 4 semi-structured interviews with community health workers, 9 semi-structured interviews with healthcare professionals; and 16 focus group discussions with the general adult population. Data were collected between June and September 2017. A content thematic analysis approach was used to analyse the data. The results of this study showed that rfMDA was accepted due to awareness about the intervention, experience of a previous similar programme, the MDA campaign, and due to favourable perceptions built on the believe that rfMDA would help to prevent, treat and eliminate malaria in the community. Perceived barriers to rfMDA include lack of access to accurate information, reluctance to take a pregnancy test, concern on drug adverse reactions, and reluctance to take antimalarial drugs without any symptom. In conclusion, the community found rfMDA acceptable for malaria intervention. But more community engagement is needed to foster community involvement and self-appropriation of the malaria programme elimination.

consent and assent after a research protocol was reviewed and approved by CISM's institutional ethics committee (CIBS-CISM) and the Mozambican Ministry of Health National Bioethics Committee. The informed consent signed by the participants stated that: "data will only be available to the study team", and the protocol stablished that all information will be confidential, and no data from the data collection forms, nor from audio files will be accessible to anyone outside of CISM. Given this statement approved by the two IRBs, data from this study is available upon request to these institutional review boards: CISM's institutional ethics committee (sozinho.acacio@manhica.net) or the Mozambican Ministry of Health National Bioethics Committee (jflschwalbach@gmail.com) for researchers who meet the criteria for access to confidential data.

**Funding:** The study reported in this paper was part of the Magude project. The Magude project (NCT02914145) was funded by the Bill and Melinda Gates Foundation and Obra Social "la Caixa" Partnership for the Elimination of Malaria in Southern Mozambique (OPP1115265). The Magude project was implemented by the Centro de Investigação em Saúde da Manhiça (CISM). CISM is supported by the Government of Mozambique and the Spanish Agency for International Development (AECID) for core funding, but the study reported in this paper did not receive any funding from the Government of Mozambique nor AECID, it was fully and solely funded by OPP1115265. The funders had no role in study design, data collection and data analysis, decision to publish, or preparation of the manuscript.

**Competing interests:** The authors have read the journal's policy and have the following competing interest to declare: CEC, BG, JML, HD, HN, IM, PA, KM, NT and FS are paid employees of the Centro de Investigação de Manhiça (CISM), which is supported by the Government of Mozambique and the Spanish Agency for International Development (AECID). The salary of CEC, BG, JML, HD, HN, IM and PA were partially or fully covered through the Magude MALTEM project, which is co-funded by the Bill and Melinda Gates Foundation and Obra Social "la Caixa". However, this does not alter our adherence to PLOS ONE policies on sharing data and materials. There are no patents, products in development or marketed products associated with this research to declare.

## Introduction

Mozambique is one of the sub-Saharan countries that has made significant progress toward malaria elimination [1, 2]. However, the country is still considered one of the 6 countries with the highest malaria burden in the world, contributing with 4% of worldwide malaria cases in 2018 [2]. Several strategies have been implemented in the country to accelerate malaria elimination in southern Mozambique [3]. These strategies include increasing the coverage of long-lasting insecticidal nets (LLINs), yearly rounds of universal indoors residual spraying (IRS), and improvement of case management and surveillance system throughout the country [3–5]. These strategies are among the recommended tools of the World Health Organization (WHO) Global Technical Strategy (GTS) for Malaria 2016–2030 [6].

Magude district, in particular, has been benefiting from a project led by the Manhiça Health Research Centre (CISM) since 2015, which aims to eliminate malaria. The project consisted of implementation of a comprehensive mixed interventions that included LLINs, IRS and four rounds of mass drug administration (MDA) to all the eligible members of the population of Magude between 2015 and 2017 using the half-life drug dihydroartemisinin-piperaquine (DHAp) [5, 7]. These interventions were implemented following different assessment and baseline studies on malaria elimination in the district [8–10] that informed the perceptions of the community before and during the implementation of the project.

Some factors influenced the implementation of malaria elimination interventions in Magude district, including refusal of IRS and LLINs use [9], absenteeism of the household head which compromised the decision-making in participation in the MDA campaign, and fear of DHAp and adverse reactions [7]. Notwithstanding these constraints, the implementation of the comprehensive mixed intervention has resulted in a substantial reduction of malaria cases in Magude district [5].

Although there is significant reduction in malaria cases in the district, elimination of malaria has not yet been achieved. Hence, the WHO recommendation of reactive epidemiological surveillance as an intervention suitable to the late stages of the fight towards malaria elimination [11]. In this context, a reactive focal mass drug administration (rfMDA) was implemented in Magude district, southern Mozambique, from July 2017 to January 2020 to maintain the gains and prevent an upsurge of malaria transmission after MDA.

rfMDA consisted of following up all passively detected malaria cases at health facilities and community health workers to their households and administering the antimalarial drug DHAp to all household members and neighbours. When a household was visited, the fieldworkers explained the reasons of the visit; enrolled the household members to the study through informed consent forms; administered electronic questionnaires to all household members gathering sociodemographic and malaria risk and prevention information; evaluated each household member's eligibility to be administered DHAp, which included pregnancy testing to consenting women of reproductive age and malaria rapid diagnostic testing to all eligible members of the households; and administrated DHAp according to each member's age. The administration of DHAp followed the same procedures used in MDA in the same district [8, 5, 7]. The implementation of rfMDA strategy was complemented by a community engagement campaign encouraging the population to seek healthcare upon the presentation of fever and to adhere to this reactive surveillance intervention.

This study analysed acceptability and perceived barriers to reactive focal mass drug administration (rfMDA) among community members exposed to community engagement campaigns and malaria elimination interventions in Magude district.

## Methods

### Study setting

The study was carried out in the rural district of Magude, located in the northwest of Maputo province, southern Mozambique. In 2017, the national statistics institute's (INE) census counted 63,691inhabitants and 14,583 households in the district [12], distributed in 5 Administrative Posts: Magude village, Motaze, Mahele, Panjane and Mapulanguene [13], and the study covered all these 5 Administrative Posts. There are 9 rural health facilities, 1 referral health centre and 27 community health workers (CHWs) throughout the district [14]. CHWs provide diagnosis and treatment of malaria and other diseases, such as diarrhoea, pneumonia and refer patients with signs of sickness requiring high medical attention [15]. Both health providers and community health workers engage in community sensitization about malaria using a social behaviour change communication approach included in the Plan of the National Malaria Control Program (NMCP) [16]. The level of malaria in the district was considered moderate before de elimination program, with about 200 cases per 1000 prior to MDA [14]. The district has been exposed to malaria prevention strategies, such as malaria case management using artemether-lumefantrine, vector control, IRS and the population has been exposed to several malaria research activities before and after the Magude project [5, 8].

### Study design

This formative qualitative study assessed acceptability and perceived barriers to the reactive surveillance strategy rfMDA among community members exposed to community engagement campaigns and malaria elimination interventions. The study was undertaken between June and September 2017, before the start of the reactive surveillance intervention and continued during the first two months after the start of the intervention in July 2017.

### Sample strategy and sample size

A purposive sampling was performed to select individual members representing different groups in the community. These groups included adult household heads ($\geq$ 18 years old), adult women of reproductive age (18–49 years old), female adolescents (12–17 years old), adult members of the community ($\geq$ 18 years old) and community leaders ($\geq$ 18 years old). The same strategy was used to select adult general population ($\geq$ 18 years old) who composed focus group discussions (FGD). These participants were selected to capture the view and the lay perspective, as well as mapping the barriers with regards to reactive focal mass drug administration. A total of 69 participants of different community groups, comprising individual semi-structured interviews, and 157 participants of the general population, who participated in FGDs, were included in the study (Table 1).

The study also included healthcare providers who were engaged in malaria campaigns and malaria elimination interventions. A purposive sampling was used to select 9 healthcare professionals and 4 CHWs in all the study settings (Table 1). Health professionals were working in the health facilities located in the same communities where the study took place. The community health workers also worked in the same communities in coordination with the local health facilities.

The study sample size did not cover all participants in all study sites due to unequal distribution of the study participants' categories and several constraints to accessing the eligible participants. Health professionals and CHWs were not included in some study sites because they were not in all selected communities, and some community leaders were absent during the data collection. Additionally, it was not possible to include adolescents from Mahele and

**Table 1. Study sample size.**

| Study setting | Individual semi-structured interviews | | | | | | | | FGDs (n = 16) with general population | | |
| --- | --- | --- | --- | --- | --- | --- | --- | --- | --- | --- | --- |
| | Household head | Women of Reproductive age | Adolescent | Member of the community | Community leader | Health professionals | CHWs | Total | Men | Women | Total |
| Magude village | 1 | 1 | 5 | 6 | 6 | 5 | 1 | 25 | 8 | 37 | 45 |
| Motaze | 4 | 3 | 2 | 4 | 0 | 0 | 0 | 13 | 1 | 16 | 17 |
| Mahele | 1 | 3 | 0 | 0 | 1 | 2 | 0 | 7 | 13 | 20 | 33 |
| Panjane | 2 | 1 | 2 | 5 | 2 | 1 | 2 | 15 | 7 | 11 | 18 |
| Mapulanguene | 1 | 2 | 0 | 3 | 1 | 1 | 1 | 9 | 16 | 28 | 44 |
| Total | 9 | 10 | 9 | 18 | 10 | 9 | 4 | 69 | 45 | 112 | 157 |

Mapulanguene because the study took place during the school season, and the eligible participants were at schools in different districts. Furthermore, members of the community from Mahele were not included in the sample of semi-structured interviews because they were unavailable due to their agricultural activities. In addition, the lack of accessibility at the selected study sites during the rainy season constrained access to the eligible participants.

## Data collection

Semi-structured interviews (SSI) and focus group discussions (FGDs) were used to collect data. Individual SSI were administered to household heads, women of reproductive age, adolescents, members of the community, community leaders, healthcare professionals and community health workers; while FGDs were used to collect data from the adult general population. The size of each FGD varied between 8 and 12 members, and each FGD lasted between 60 and 80 minutes. Data collection guides for both SSI and FGDs were designed to capture perceptions of rfMDA, acceptability of the procedures of rfMDA and the reasons for its acceptability, and barriers that could emerge during the implementation of rfMDA. Guides were prepared in Portuguese, and a pilot test was performed in the local language Changana before the beginning of data collection. Based on the pilot test, the guides were refined. SSI were conducted in both Portuguese and Changana, depending on the language preference of the participants, while all FGDs were conducted in Changana. The interviewers, who are fluent in Portuguese and Changana, were trained to conduct SSI and facilitate FGDs. All interviews and FGDs were digitally recorded, and later independently transcribed in Portuguese. The research team controlled the quality and accuracy of the transcriptions.

## Data analysis tools

A content thematic analysis approach was used to analyse the data of SSI and FGD. First, data management was conducted using Nvivo 12 (QRS International Pty. Ltd.), a qualitative package for qualitative data analysis, following designed generic outline nodes representing the codding structure. Themes and subthemes emerging from the data were critically discussed until a consensus of the researchers was reached. The final themes were: awareness and acceptability of reactive focal mass drug administration, acceptability of the procedures used in rfMDA and barriers to rfMDA.

## Ethical considerations

The study was approved by CISM's institutional ethics committee (CIBS-CISM) and the Mozambican Ministry of Health National Bioethics Committee, and it was registered as

protocol number Ref:146/2017. All participants received detailed information about the study objectives. A written informed consent was obtained from all participants prior their participation in the study. The study obtained a written informed consent from all parents or guardians of the young adolescents (12–17 years old) included in the study. Additionally, an assent was sought from all young adolescents that participated in this study. Participants were assured about their anonymity and confidentiality throughout the research process. Thus, all participants names were not recorded, and all informed consents, digital records and databases were securely stored at a secure server of CISM.

## Results

The participants of this study included different community groups, general population of the community, healthcare professionals and community health workers. Tables 2 and 3 summarise the characteristics of participants per community group and among the general population who participated in semi-structured interviews and in focus group discussion respectively. The majority of participants were married or living with a partner, had primary school and worked as famers.

Table 4 presents the characteristics of healthcare professionals and community health workers. The majority of participants had secondary school. Almost all healthcare professionals had specialised training in primary healthcare and working as maternal and child health nursing, general nursing, technician of preventive medicine and assistant of service, while community health workers had not any specialised training.

**Table 2. Sociodemographic characteristics of participants per community group.**

| Characteristics of participants | Community leaders (n = 10) n(%) | Household head (n = 9) n(%) | Women of reproductive age (n = 10) n(%) | Adolescents (n = 9) n(%) | Members of the community (n = 18) n(%) |
|---|---|---|---|---|---|
| **Sex** | | | | | |
| Male | 10 (100) | 7 (77.8) | 0 (0) | 0 (0) | 3 (16.7) |
| Female | 0 (0) | 2 (22.2) | 10 (100) | 9 (100) | 15 (83.3) |
| **Educational level** | | | | | |
| None | 1 (10) | 3 (33.3) | 1 (10) | 0 (0) | 2 (11.1) |
| Primary school | 9 (90) | 6 (66.7) | 6 (60) | 7 (77.8) | 16 (88.9) |
| Secondary Education | 0 (0) | 0 (0) | 3 (30) | 2 (22.2) | 0 (0) |
| **Marital Status** | | | | | |
| Single | 0 (0) | 0 (0) | 3 (30) | 7 (77.8) | 1 (5.6) |
| Married or living with a partner | 9 (90) | 9 (100) | 7 (70) | 2 (22.2) | 17 (94.4) |
| Widowhood | 1 (10) | 0 (0) | 0 (0) | 0 (0) | 0 (0) |
| **Occupation** | | | | | |
| Farmer | 10 (100) | 7 (77.8) | 8 (80) | 2 (22.2) | 14 (77.8) |
| Salesperson | 0 (0) | 1 (11.1) | 0 (0) | 0 (0) | 1 (5.6) |
| Security | 0 (0) | 1 (11.1) | 0 (0) | 0 (0) | 0 (0) |
| Housewife | 0 (0) | 0 (0) | 1 (10) | 1 (11.1) | 2 (11.1) |
| Traditional healer | 0 (0) | 0 (0) | 0 (0) | 0 (0) | 1 (5.6) |
| Student | 0 (0) | 0 (0) | 1 (10) | 6 (66.7) | 0 (0) |
| **Religion** | | | | | |
| Atheism | 1 (10) | 2 (22.2) | 1 (10) | 0 (0) | 0 (0) |
| Christianity | 9 (90) | 7 (77.8) | 9 (90) | 9 (9) | 16 (88.9) |
| Animism | 0 (0) | 0 (0) | 0 (0) | 0 (0) | 2 (11.1) |

Table 3. Sociodemographic characteristics of focus group discussion participants.

| Characteristics of participants | Frequency | % |
|---|---|---|
| **Sex** | | |
| Male | 45 | 28.7 |
| Female | 112 | 71.3 |
| Education level | | |
| None | 51 | 32.5 |
| Primary | 87 | 55.4 |
| Secondary | 19 | 12.1 |
| **Marital Status** | | |
| Single | 21 | 13.4 |
| Married or living with a partner | 118 | 75.2 |
| Widow/Widower | 18 | 11.5 |
| **Occupation** | | |
| Farmer | 123 | 78.3 |
| Labourer | 14 | 8.9 |
| Salesperson | 7 | 4.5 |
| Housewife | 5 | 3.2 |
| Students | 3 | 1.9 |
| Traditional healer | 5 | 3.2 |
| **Religion** | | |
| Atheism | 24 | 15.3 |
| Christian | 125 | 79.6 |
| Animist | 8 | 5.1 |

## Awareness and acceptability of reactive focal mass drug administration

**Awareness of reactive focal mass drug administration.** Participants of this study were aware about the rfMDA programme that was taking place in the community, and they had participated in the previous MDA campaign. Participants received information about rfMDA from community leaders, community meetings, radio, fieldworkers, neighbours and health-care professionals after visiting a health facility and being tested for malaria. All participants of

Table 4. Sociodemographic characteristics of healthcare professionals and community health workers.

| Characteristics of participants | Healthcare professionals (n = 9) n(%) | Community health workers (n = 4) n(%) |
|---|---|---|
| Sex | | |
| Male | 4 (44.4) | 2 (50) |
| Female | 5 (55.6) | 2 (50) |
| Education level | | |
| Primary | 0 (0) | 3 (75) |
| Secondary | 8 (88.9) | 1 (25) |
| High Education | 1 (11.1) | 0 (0) |
| Marital Status | | |
| Single | 6 (66.7) | 0 (0) |
| Married/living with a partner | 3 (33.3) | 3 (75) |
| Widow | 0 (0) | 1 (25) |
| Religion | | |
| Atheism | 1 (11.1) | 1 (25) |
| Christian | 8 (88.9) | 3 (75) |

different community groups said that the objective of rfMDA was to treat, cure and eliminate malaria. They viewed rfMDA as important to their families and communities because it helped to diagnose, treat and prevent malaria, which they perceived as a problem in the community. Additionally, participants also perceived that since the beginning of MDA and rfMDA programmes, their health status had improved, malaria cases had decreased, and they believed that these programmes cured malaria. One of the participants said:

"*I think it is good because before this project* [rfMDA] *started, when my son and I got sick, I knew beforehand that the other one would also get sick quickly, so I had to get money urgently and go back to the hospital, but since the distribution of the pills, my children and I have not got sick until today*" (FGD 04, general population, Mahele).

Participants had experienced the rfMDA program, and they said that it consisted of diagnosing, treatment and prevention of all members of the family, as one of the participants expressed his opinion as follows:

"*Even myself I got sick with malaria, they came in my house to test, no one else was diagnosed with malaria, but everyone was given pills even without having malaria. They didn't give me more pills because I was taking pills*" (FGD 15, general population, Motaze).

**Acceptability of malaria reactive focal mass administration.** All participants of different groups of the community regardless their place of residence, accepted and welcomed the rfMDA programme because they perceived that it saved people from dying from malaria, eliminated malaria in the community and helped to improve their health status. Some participants expressed their views as it follows.

"*The community accepts* [rfMDA] *because they are seeing that they have no other way to prevent the outbreak of malaria or eliminate malaria because malaria kills. It is imperative that they accept and comply with the recommendations so that we can eliminate malaria*" (FGD 01, general population, Panjane).

"*I accept because I see that the fieldworkers follow us from hospital to our homes because of this malaria disease. When they do follow up it allows everyone to be diagnosed, including those who do not like to go to hospital, and so one can fight and eliminate this disease* [malaria]" (FGD 09, general population, Motaze).

Furthermore, all participants accepted rfMDA because it is based on home treatment, which reduced the cost of transport to the health facility, and helped people who are lazy to go to the health facility when they have malaria symptoms and those who live far from the health facility. However, in one of the FGDs with general population, participants reported accepting rfMDA because they were following norms from the health facility, and they perceived that if they do not accept malaria treatment, they might experience difficulties in the future malaria treatment at the health facility.

"*Researcher*: *Thank you very much. Do you think it is important that we distribute pills in the districts*?

*Participant*: *It is very important, it helps us with diseases, even the persons who are lazy to go to the hospital when they have malaria symptoms, they end up taking it, because the pills go to their house*" (SSI 03, community leader, Magude village).

"*Haaa. . . we accept because those are the norms and you must comply with. If you don't accept to be cured, when you go to hospital (. . .) while you have malaria, they* [healthcare professionals] *will say that you are not sick with malaria because you didn't accept this treatment* [rfMDA]. *They will say that you are happy when people die in the community, and that when you get malaria you will contaminate everyone. So, we accept that when one person from the household gets sick, the fieldworkers come to test the rest of the household members so that everyone is protected*" (FGD 01, general population, Panjane).

Several participants of different members of the community assumed that everybody would accept to participate in the rfMDA programme because people were aware of the severity of malaria including its death consequences, and also because they had experience of the benefits of the previous similar campaign against malaria (MDA). They also added that people were aware that they had common consensus regarding malaria. This consensus consisted on the idea that malaria was a problem of all members of the community, and therefore, they had to fight against it; and they viewed rfMDA programme as a vehicle which helps to eliminate it.

"*Everybody will adhere to the programme because uhm, malaria kills. And at that time before these pills existed others died because of this disease (. . .). Because what happens is that when people get malaria today, tomorrow they wake up well, it attacks them the day after tomorrow, the next day they wake up well, when malaria is rising and then it gets to the point that they don't even wake up and then go to hospital when it has risen, the person is already losing his life by then. But soon after those pills arrived, we escaped, I still haven't heard that anyone has died of malaria now, since we have been taking those pills. Now even if they go around the houses giving us pills there is no one who will deny; people will accept*" (FGD 09, general population, Motaze).

## Acceptability of the procedures used in reactive focal mass drug administration

The rfMDA consisted of following up all patients tested for malaria at the health facility or by community health workers. Fieldworkers followed the patients to their homes, performed malaria and pregnancy tests, and treated all eligible household members and the surrounding neighbours. This theme analyses community acceptability of these procedures.

### From health facility to home treatment

All participants accepted and welcomed the procedure of following up patients from the health facility to their homes. Participants perceived that this procedure would prevent high transport costs from home to the health facility, it would enable them to know the number of people infected by malaria at the household, and it could contribute to eliminate malaria and prevent death from it. Moreover, participants also perceived that a visit from the health facility showed an interest of the healthcare professionals about patients that tested positive to malaria. The following exerts present participants' views who had experience of rfMDA.

"*We used to die a lot from malaria, because when the person was shaking and could not go to hospital, and ended up dying inside the house (. . .) because many people do not have possibilities to take the sick person to hospital. Now, treating the disease* [malaria] *at home, this will decrease malaria and avoid deaths from malaria*" (FGD 09, general population, Motaze).

"*Participant 2*: [Fieldworkers] *came to my house because I went to the hospital and tested for malaria. They came to my house to visit me. They said they were going to visit other people who had also been diagnosed with malaria in hospital. So, they visited me up to two to three times. I thank them for the visit since they are visiting me, they want to know if I am better or not.*

*Participant 5*: *They are good visits, because they are visiting us after we go to the hospital to know how we are doing, it is good like this when healthcare professionals visit us*" (FGD 05, general population, Magude village).

Healthcare professionals, in particular, hypothesized that communities would accept receiving fieldworkers from the health facility because the procedure will prevent many patients from having to go to the health facility, where they often spend a long time to be treated or attended. In addition, healthcare professionals viewed the procedure as an opportunity to visit communities; and a such visiting could represent the commitment of the healthcare professionals with the communities and it could strengthen the relationship between them; while others perceived it as an opportunity to identify other patients who could have malaria symptoms and monitor those who have already tested positive to malaria.

"*It is a welcome activity because, firstly, when they receive a visit from healthcare professionals, the community feel valued because they know the healthcare professionals go out from the health units to the community to find out about the health situation of that community. For the communities, the visit shows some interest of healthcare professionals to the community. First, we gain that trust with our community as an institution and second, I can say that we manage to detect the possible cases* [of malaria] *that may be emerging and at some points hidden in the community*" (SSI 04, healthcare professional, Mahele health facility).

"*Following participants who test positive for malaria is a good activity, because when we go to the house, after we have tested a member, we can see if that member who tested positive for malaria is or is not complying with the medication. But, also at home there might be another member with malaria, so when we go there* [in the household], *we test, we will know how many people have malaria*" (SSI 05, healthcare professional, Mapulanguene health facility).

**Acceptability of malaria testing at home.** Several participants accepted to be tested for malaria at home because they perceived that testing was a way of diagnosing malaria, which a lot of the times can be asymptomatic. In addition, participants said that the home testing enabled to diagnose other diseases that people might not know.

"*I accept to do the test because when someone appears who was bitten by mosquitoes, they go to the hospital, then they are able to follow up on that case, they go to the house of the person who was detected with malaria, test the people from home, medicate so that they don't get sick. They do that because that person who was detected malaria and it can be the case that the mosquito contaminates the other people, but there can also be people with malaria in that household who have not yet gone to the hospital*" (SSI 05, household head, Motaze).

Participants also perceived that testing was the only guarantee to know their health status and to comply with the prescribed medication. They said that they wished to be tested to know if they had malaria or not, and only thereafter they would be sure about the disease they are suffering from and take the prescribed pills.

"*If the fieldworkers come to my house and they don't test me, I don't feel happy. I want them to test me until they tell us that we don't have malaria, only then will we feel happy, because even if you go to the hospital and then arrive with the child when he is sick, if they don't test him and then take any pills and give to him without testing him, he won't feel comfortable. If the child takes the pills and the next day he doesn't get better, he will say that it is because they didn't do any analysis, maybe it's malaria, you don't feel happy*" (FGD 09, general population, Motaze).

**Acceptability of including neighbours in malaria treatment.** Participants were asked if they would accept malaria treatment after their neighbours were tested positive to malaria. All participants said that they would accepted malaria treatment if their neighbours tested positive to malaria, even if none of their household members was tested positive to malaria. This acceptability derived from the fact that participants perceived that malaria was transmissible, and for that reason, including neighbours in malaria treatment would prevent others from getting the disease.

"Participant 3: I accept because I will not only prevent the people in my house, but also the neighbours (...). This activity of fighting malaria, eliminating malaria from neighbour to neighbour is good because we will all be free from malaria.

Participant 1: In my opinion, I see that it is very good when the fieldworkers come to test me for malaria and also test the people at home and the neighbours, because it may happen that the mosquito that bit me comes back to bite the people here at home and the neighbours. The mosquito can enter in the house of the immediate neighbours.

*Participant 5: Once I have been infected with malaria it may happen that the neighbours are also infected because the mosquito bites here, comes out and bites the neighbours. I see these activities are very important to prevent malaria*" (FGD 13, general population, Mapulanguene).

**Acceptability of pregnancy testing at home.** All participants of different groups said that they would accept pregnancy testing at home. Participants were aware that a pregnant woman should not take malaria pills. In addition, participants said that women of reproductive age might not know if they are pregnant or not, and the test would help to disclose the status of the women before administration of the pills.

"*Participant 3: We accept the pregnancy test because the fieldworker will be following the norm "that you cannot give pills if I am pregnant, it may happen that I say I am not pregnant, while I am, I want to undo the pregnancy to relieve myself". So, I don't see a problem in this issue of taking pregnancy test to know if you are pregnant or not. Also, even if the person has not spoken, it is necessary that they first be tested to know if they are pregnant or not, because it can happen that they say they are not, while they are, they give pills and the pregnancy undoes itself.*

*Participant 5: In a household there can be girls, one of them can be pregnant and no one in the house knows, she got pregnant and so on, it's not official* [refers to a pregnancy contracted from a man not known to the family members and who has not gone through some ceremony of making the relationship official] *so, no, the culprit will not be the fieldworker, because they also did not know of the existence of the pregnancy.*

*Participant 7: It is also not correct that a girl is pregnant and takes the pills. If the girl is pregnant and after taking the pills the pregnancy falls apart, it would be the fault of the fieldworker*" (FGD 15, general population, Motaze).

Both women of reproductive age and adolescents accepted to be tested, and they also knew the importance of pregnancy test before the administration of the malaria pills. They believed that if a pregnant woman took the malaria pills she could suffer abortion. They perceived the pregnancy test as a way of preventing abortion. In addition, women of reproductive age and members of the community said that they were "*pleased*" to undergo a pregnancy test because it enabled them to uncover the pregnancy.

"*We do pregnancy tests for women because it can happen that they give pills while she is not well, if they give pills while she is pregnant, she can have complications or lose that pregnancy here at home, the fieldworker who gave the pills will be guilty*" (. . .) (SSI 05, woman of reproductive age, Motaze).

"*The test is very good because you can be pregnant without knowing. The first time I was tested I was breastfeeding my baby and I didn't know that I was already pregnant. When they did the test, they found out that I was pregnant, but I didn't even know, they did me a big favour because even my husband didn't know; the pregnancy was hidden, the child was sucking dirt (. . .). If it hadn't been for the test, I would only realise that I was not well when the belly was already big, so the test was very important*" (SS 02, woman, member of the community, Magude village).

Moreover, household heads, both women and men, and community leaders mentioned that they accepted pregnancy test to their wives and female adolescent as they acknowledged that they might not know if they were pregnant or not. In addition, they viewed a pregnancy test as "*good*" because it helped to uncover several diseases, and it enabled pregnant women to seek the health facility early on for treatment and follow-up of the pregnancy.

"*The pregnancy test is important because if the person is tested they* [fieldworkers] *can find many other diseases; if they find diseases, the doctors will treat those diseases that she has. The person is tested because it may happen that she is pregnant while she has malaria, the child may get it from inside the mother* [in pregnancy]. *When the woman is tested, various diseases will manifest then, so that both mother and child will be treated*" (SSI 10, household head, Motaze).

"*Participant 1*: *When they test us and find out that we are not pregnant we are happy because we are breastfeeding.*

*Participant 3*: *Testing girls for pregnancy does not pose any problems because they have grown up. For us mothers, if it is me, finding my daughter in this state* [pregnant], *for me it is a help because I live with her without knowing. It happened to me, I want to be honest, I sent my daughter to school without knowing that she was pregnant. The school sent her back home because she was pregnant, but if I had known before, I wouldn't have sent her to school.*

*Participant 5*: *I don't see any problem in testing my daughters because if you find out that my daughter is pregnant, and tell me I will have information or tell her in secret, she will come to know that she is pregnant (. . .); there is no problem (. . .)*" (FGD 07, general population, Mapulanguene).

**Acceptability to take malaria pills at home.**   Participants accepted to take malaria pills at home even when they were not sick with malaria as they perceived that pills prevented malaria to the members of the family and community members, which in turn prevents people to go to the health facility often because they lived far from the health facility. In addition, a

community leader stated that since the start of the mass drug administration, he has witnessed a reduction in malaria cases. The same participant also said that the community had learned from previous experiences, such as MDA, that malaria pills protect people from diseases.

"*I accept taking tablets even without malaria. Even if field workers leave my neighbour's house after giving pills, come here at home, we all have a duty to accept, because since we started taking pills in 2016 until now we have seen a reduction in malaria. So, we should not refuse, we have to accept taking tablets to prevent malaria*" (SSI 09, community leader, Magude village).

Regarding the easiest group to accept malaria pills, participants mentioned young and adult women, adult men, elders, community leaders and all people with the experience of malaria disease and those who were not willing their family members to get it.

"*Neither our ladies' group nor the gentlemen's group can refuse, because when you start to get sick, no one is happy about it, we rejoice when our children and we adults are in good health. Therefore, we cannot refuse* [to take pills]" (FGD 04, general population, Mahele).

"*I think the group of mothers are the ones who understand the most, because they have younger children. They quickly understand why they prevent themselves and their child's health. They usually follow the healthcare programmes. The elderly also easily accepts to take the pills. In general, adults will accept because they comply with one thing and another that is said. When you speak, they feel firm in your words and you make sure that you also do it in your house, they like it*" (SSI 04, community health worker, Mapulanguene village).

### Barriers to reactive focal mass drug administration

Questioned on the main barriers to the reactive focal mass drug administration, the different community groups said that there were some barriers regarding the ongoing implementation of rfMDA. They predicted that not everybody would accept to be tested and some community members might insult or mistreat the fieldworkers because each member has its own way of thinking. Additionally, participants said that some household heads might not allow fieldworkers to enter in the house and treat the members of the family, or fieldworkers might be poorly treated, while others pointed out issues related to the absence of some or all members of the household. For the participants, these barriers could hinder the rfMDA programme.

"*It is possible that the person you are going to meet in some household will insult you; he may say: go back with that job of yours (. . .). Other people may make jokes and talk a lot of nonsense (. . .)*" (SSI 05, community leader, Magude village).

"*The only barriers they* [fieldworkers] *can find are like arriving at a house and not finding anyone. After sometimes, this family may get sick while people* [fieldworkers] *have already passed (. . .)*" (SSI 17, member of the community, man, Panjane).

**Barriers to home testing for malaria.** Regarding the barriers to home testing, participants mentioned some barriers, such as the repeated pricks to collect blood samples and difficulties to collect blood samples among children because participants perceived that the blood of the child would finish as children do not have a lot of blood. In addition, it was also mentioned that some household heads might not accept the test for themselves and their family members due to lack of awareness about the malaria test.

"*Difficulties may exist when fieldworkers prick children and the blood doesn't come out, or when they prick someone and the blood doesn't come out; when they insist and prick up to*

*three times on the same finger the person starts to feel pain. And, when it's a child, if they prick several times the blood will finish because the child still doesn't have much blood*" (FGD 07, general population, Mapulanguene).

"*What might be a hindrance to the activity is if the head of the household does not accept the malaria test for himself and his household members because he might not think it is important (. . .). If the householder refuses, it will not be possible to do the malaria test*" (SSI 04, member of the community, man, Magude village).

**Barriers to pregnancy test.**   Participants presented several barriers regarding pregnancy test, which included, management of positive pregnancy test disclosure specially when the women's husbands work far from home, existence of difficult groups to preform pregnancy tests, perceptions about who should perform a pregnancy test in women, as well as, the fear of family problems.

Participants agreed that it would be difficult to test and manage pregnancy test results among women whose husbands work and live in South Africa. In fact, among the male participants, particularly the ones working far from their homes, raised a concern regarding the disclosure of the pregnancy test result. The concern was that the disclosure of pregnancy in their absence could create worries as the community would be the first to know, and they might not certify if their wives were faithful. Thus, they requested that the disclosure of the pregnancy test should be a secret.

"*There will be problems in my house with my sister-in-law because her husband is not in, he went to South Africa. So, if the fieldworkers find out that she is 2 months pregnant while her husband has long travelled to South Africa, we need to have a good talk with her. But if it is my daughter who is pregnant, there is no problem. You can tell me*" (FGD 07, general population, Mapulanguene).

"*Participant 3: Regarding the difficulties of pregnancy testing for women, we request that your fieldworkers who will be distributing pills, have confidentiality because from my wife's side, I work and stay a long time on duty, I end up staying 2 months without coming back. I may think that my wife has nothing* [pregnancy] *while she is pregnant. So, if there is a leak that my wife is pregnant and I don't know, nor have I seen; excuse me, but we need to be clear, because I will no longer know if that pregnancy is mine or not. Your fieldworkers should have confidentiality; (. . .) you didn't come to destroy our homes, you came to help us, so we ask for confidentiality when it is proven that women are pregnant.*

*Participant 5: I agree with what the colleague said. It would be good if fieldworkers could test and say how many months of the pregnancy: one or 2 months; because I can stay in South Africa (. . .) 3 or more months working outside home and, the fieldworkers find out that my wife is 2 months pregnant, but I have been outside home for more than 3 months (. . .). Then when they find out that she is pregnant, the fieldworkers cannot talk in the community because they have not come to destroy our homes*" (FGD 08, general population, Mapulanguene).

Regarding the difficult groups to perform pregnancy test, both adult women and men, community health workers and community leaders mentioned adolescents. They predicted that adolescents may refuse the pregnancy test at home due to fear of their parents, because if they are tested positive, their parents would know that they are pregnant and this can be a family issue as they might be hiding the pregnancy.

"*The ones who usually deny pregnancy test are the girls. Since the test will be done at home, they know that if they test me here where the breast is, she will find out that I am pregnant*" (FGD 09, general population, Mahele).

"*Girls tend to hesitate to take the pregnancy test. They say they are not pregnant while they are hiding*" [the pregnancy] (SSI 10, community leader, Mapulanguene).

Adolescents, however, said that they were not afraid of pregnancy tests. They added that who had to decide about pregnancy testing for them were their parents. They perceived that their parents may not allow them to do pregnancy testing due to social norms. They explained that if they are found pregnant they had to inform their mothers, and not their fathers or the mother and the father at the same time.

"*Mums might not accept their daughters taking the pregnancy test because if me and my parents, mummy and daddy are sitting in the same place, no matter how much something forbidden happens to me* [menstruation or pregnancy] *I can't tell my dad. I have to go and tell my mummy because I don't know anything yet, I'm underage, they tell me to do this, this and this, and I say 'that's fine thank you'. Then mummy might not accept that I do pregnancy test in front of my daddy because he will know the results immediately*" (SSI 01, adolescent, Magude village).

Women of reproductive age said that some household heads might not accept their wives to perform a pregnancy test because men perceived that a pregnancy test must be performed by a woman and not by a man fieldworker. Participants added that some women might refuse pregnancy test due to fear of violence of their husbands.

"*The group that might not allow women to do the pregnancy test are men, because they think that a man has no right to test pregnancy on a woman, only a woman can test pregnancy on another woman*" (SSI 17, member of the community, woman, Panjane).

"*Women may refuse pregnancy testing for fear of violence of their husbands; this can happen. Some men may be violent to their wives if they accept the test without their consent*" (FGD 10, general population, Magude village).

However, household heads said that women of reproductive age do not like to do pregnancy tests at home because they said that if they wanted to know about their reproductive health, they would go to the hospital.

"*There are many women* [of sexually reproductive age] *who do not like to take a pregnancy test. When they are talking on the street, they say that 'testing people is not good, because if I want to have a baby, I know the way to the hospital, I know how to do it, testing people is not good'. (...). It has been more the women who deny the pregnancy test because they say that they know where to get help, which is in the hospital*" (SSI 04, household head, Motaze).

Participants also said that some women might deny pregnancy test due to fear of pregnancy disclosure within the family. Additionally, they said that some women might also make use of pills distributed to prevent malaria to do the abortion of unwanted pregnancy, as they are already aware that malaria pills may interfere with the pregnancy.

"*Other women may refuse to do the pregnancy test if they know they are pregnant and they did not want to* [unwanted pregnancy], *and they may want to take the pills without testing to*

*take advantage of the pregnancy* [have an abortion]. . . *because in the other malaria campaign* [MDA] it was *said that if you take pills when you are pregnant, the pregnancy will come out* [you can have an abortion]. *But, other women can refuse to be tested because it can be found out that they are pregnant. . . . Our daughters may not know that they are pregnant, but after the test they will know and we will also find out and ask them about the pregnancy*" (FGD 09, general population, Motaze).

**Barriers to the administration of malaria pills at home.** All participants pointed out some barriers that can hinder the uptake of malaria pills. These barriers included people's perception and habits about when to take pills, side effects, lack of compliance on the dosage, lack of decision-making by the household head, conflict of prescription between the recommended malaria pills and local traditional medicines, lack of adequate information, and existence of groups who can refuse to take pills.

Regarding people's perceptions about when to take pills, healthcare professionals mentioned that most members of the community perceived pills as substances to be taken when they are sick, and it would be challenging to request people to take malaria pills while they were not feeling sick.

"*I think that there will be some barriers because our communities, the characteristic of our communities, is to take some pills when they are sick. So, when you arrive in the community and tell people to take pills while they do not feel sick, then this ends up creating a situation that is not good for the community. So, this is the main barrier that even we as an institution, we have been facing because they only take pills when they are sick*" (SSI 04, healthcare professional, Mahele health facility).

In fact, to substantiate healthcare professionals' predictions, household heads and adolescents confirmed that they would not take malaria pills unless the test shows that they have malaria, even if their neighbours or other members of the family were tested positive to malaria.

"*(. . .) I cannot accept taking pills just because they tested and found that my neighbour had malaria while my test was negative, because they tested to know if I have malaria, and they told me that I don't have malaria; and then if they give me pills to take; that I cannot accept*" (SS 01, household head, Panjane).

"*I can't accept to take pills because I don't have malaria, even if my neighbour was detected malaria in the hospital*" (SSI 01, adolescent, Panjane).

Another barrier was regarding participants' previous experiences of malaria pill's side effects. Participants said that some people might not accept taking malaria pills because when they took in malaria pills in the previous campaign (MDA), they experienced dizziness.

"*People may not take the pills because of dizziness, because the pills cause dizziness; they make you dizzy. It happened with my grandson, he got dizzy, he was shaking after taking malaria pills in the second day. We went to the hospital and they prescribed other pills that we have to buy from the pharmacy outside, but the pharmacy was closed because it was Sunday, and it was difficult to manage the situation*" (FGD 11, general population, Magude village).

Lack of compliance with malaria pills dosage was also reported as a barrier. The discourse of participants pointed out that some people only took the pills in the first day, in the presence of the fieldworkers. But, they did not adequately take the pills in the following 2 days as they had been recommended.

"*I think that there are still difficulties in taking the pills because some people, when the fieldworkers leave those pills that they have to take in the absence of the staff, some don't take it. I can believe that some don't take it, this is because the same person. . . the same family member, whose other was tested positive, when they leave it for him to take it, he doesn't take them, and three days later he shows up at the health facility with malaria, and sometimes, when we ask if he took the medicine that fieldworkers left, and he says yes, while he simply didn't take it*" (SSI 2- healthcare professional, Magude village health facility).

"*(. . .) Most people do not take pills until finishing the dosage. They interrupt it and drink beer, but they won't get better, they will always be in hospital because they have transgressed the norms, crossed the line, and they will always get sick*" (FGD 01, general population, Panjane).

Healthcare professionals and community health workers mentioned the absence of the household head or lack of his consent as a barrier to all family members to take the pills.

"*One of the barriers would be if field workers arrive in a household where the head of household is not there, practically that person will not be attended to. Fieldworkers will not be received, they will have to wait for the head of household to authorise, then they will not be able to work*" (SSI 05, healthcare professional, Panjane health facility).

According to the participants, the intake of traditional medicines might be another barrier to home intake of the drug. Community leaders and healthcare professionals mentioned that there might be a conflict between the recommended malaria pills and local traditional healers' practices. They explained that some traditional healers may refuse malaria pills alleging that they treat it themselves. Others said that children or other people might not be allowed to take malaria pills at the same time that are taking traditional medicine prescribed by the traditional healers.

"*Another barrier would be to get to the household head, let's suppose that the head of that household is a healer, he thinks he can treat malaria, or he can only treat the person who has malaria, not those people who don't have it, he knows how to do things. He will say: no, here at home these are the rules, I treat it, no one get sick of malaria (. . .). It would be difficult to convince him because he thinks that he can treat himself, he is already a doctor, he calls himself a house doctor, it would be difficult to medicate this healer, because he thinks that he is also a professional. And he may not let the fieldworkers do their job because of some myths. You can explain to him that there is no traditional treatment for malaria, but he still has these taboos*" (SSI 05, healthcare professional, Panjane health facility).

"*For example, here in Mapulanguene* [name of administrative Post], *there are traditional healers who prescribe traditional medicine to children and other people. You may come to a family, and they can say: "today I gave traditional medicine to my son, and he/she cannot take malaria pills", you may find that*" (SSI 10, community leader, Mapulanguene).

The future possibility of getting malaria sometimes after taking the pills was mentioned as another barrier. Healthcare professionals said that some people might ask "*for how long they will get malaria after taking pills*?", and if they are aware that even taking the pills, after sometimes (approximately 6 weeks) they can still get malaria, they might not adhere to the pills.

"*One of the barriers would be, for how long will I not have malaria, for how many years? That question anybody can ask, as long as they don't have exact information about the drug, they can ask this question for how long, if it's for a short time, he or she may reject saying: 'there's. . . I don't have malaria, what's the point if after so long I'll have malaria*" (SSI 05, healthcare professional, Panjane health facility).

Indeed, Participants with previous experience of malaria pills treatment questioned the usefulness of the malaria pills because they still got sick even after taking the pills.

"*We heard that malaria will end after taking the pills, we took the pills but we still get sick with malaria*" (FGD 08, general population, Mapulanguene).

Several participants said that the main barrier would be the lack of adequate information about the importance of pills for malaria prevention. They also added that another barrier would be lack of information about how and when to take the malaria pills. Participants reported that not all fieldworkers offered adequate information before requesting people to take the pills.

"*Inform your fieldworkers who are distributing pills, in the beginning there were problems because people said that: 'I cannot take pills because we had not eaten', and we are not yet well clear in our heads. We asked that when the campaign starts, also bring food because we thought we could take pills after the meal; while it is not. We went to find out that it was a mistake of some fieldworkers. It is not everything that they tell us, that they explain clearly in the households. Some fieldworkers misrepresent the information, it is important that they come while they have clear knowledge of what they are going to do. They say that these pills can only be taken after the meal*" (FGD 08, general population, Mapulanguene).

In fact, healthcare professionals experienced the impact of this misinformation in some communities. They reported that some people refused to take pills unless it was also accompanied with some food distribution.

"*The big barrier, which is not even my opinion, but it is what I have experienced in the community, is that once I went to talk to my neighbour, I tried to convince her to take the pills, but she did not accept for the following reason, she says: "first they should give us food, they always only come to give us pills after pills, first you have to eat to be able to take pills. Why don't they give us food? They are only handing out pills*", this is one of the barriers that is common in the community*" (SSI 07, healthcare professional, Magude village health facility).

Additionally, community health workers and general population who participated in this study reported that it would be difficult to convince members of the community to take pills because the fieldworkers were outsiders, and local community health workers or members of the local communities were not involved in the campaign.

"*In the previous campaign, it would have been possible to eliminate malaria, but it was not possible because outsiders were recruited and worked in the campaign. We had problems*

*because they* [fieldworkers] *did not work with us* [local community health workers]. *So, in some households, they had difficulties because people did not accept to take pills as they did not trust those who were distributing pills*" (SSI 04, community health workers, Mapulanguene).

With regards to the groups that are resistant to take malaria pills, participants presented mixed perceptions. Participants said that young people, particularly boys and drunken people were mostly the groups that would refuse to take pills.

"*The group that refuses to take pills is the group of boys, because I have a boy who refuses to take it, he does not accept it, but we take it* [adult men and women] *(. . .). We don't succeed to convince young people to take pills. They will not take it. You can meet them here at the gate and say that you need person 'X', he will tell you that he has just left, while it's himself. The fieldworker will leave, but if I tell the fieldworker that 'that is the person you are looking for'; he can turn and kill me*" (FGD 10, general population, Magude village).

"*We drink beer, when you arrive, I will have already drunk beer, they* [fieldworkers] *give us medicine and tell us to take it, while we are already drunk. Even those pills that others say are bad, in reality, they don't make you sick, when the fieldworkers arrive they find me drunk and they tell me to take* [pills] *there in the presence of them* [fieldworkers], *so, the person gets drunk twice*" (FGD 06, general population, Magude village).

Adolescents and healthcare professionals, however, perceived that adult people working in South Africa and elderly were groups that would mostly refuse to take pills.

"*The majority, the new ones, (. . .) I'm talking about the young people, those don't have problems. I believe that a big part of the people who inhibit family members from taking pills, are adult people who work in South Africa, because they don't know where we are coming from and where we are going to. They don't get the information in the first hand, or hear it from someone; they only hear rumours, and they end up inhibiting their relatives from taking the pills*" (SSI 08, healthcare professional, Magude village health facility).

"*The elderly and fathers* [adult people] *only take pills when they want, others only take them in the first day, the next day they don't take them, and they say*: "*as soon as they* [field workers] *are gone, they won't see that we are not taking it*" *and, they leave the pills*" (SSI 10, adolescent, Panjane).

**Perceptions about ways to increase adherence to reactive focal mass drug administration.** All participants of different community groups perceived that several strategies could be used to increase community participation in rfMDA, including the need for more awareness about rfMDA, planning of the activities, access to accurate information about antimalarial pills, supervision during the administrations of the pills and improvement of attitudes of fieldworkers.

The access to accurate information was considered crucial to increase adherence to rfMDA. Thus, participants suggested more community engagement including door to door sensitization, use of entertainment activities, such as theatre for sensitization, as well as the inclusion of community leaders during the campaign and rfMDA implementation.

"*Community leaders should be informed to gather the population and inform them about the malaria campaign. They should be informed about the month and day when the fieldworkers*

*will come to the community. People should be informed about the importance of the pills and appeal to the population not to run away during the fieldworkers' visit. When the campaign starts the community leaders should be informed and they should accompany the fieldworkers because they are the ones who know the communities*" (FGD 09, general population, Motaze).

Participants also said that they were often busy with their everyday activities, and they might not be at home during the visit of the fieldworkers. So, they proposed that rfMDA activities should be well planned, people and community members should be informed beforehand about the day and time the fieldworkers will visit, and also, they should comply with the planned day. Participants perceived that this would prevent absence of the members of the household.

Several participants reported that it was important to give accurate information about antimalarial pills in advance. They explained that people should be informed about the importance of the pills, explaining its adverse effect and evaluating if some people are sick of some disease contraindicated to antimalarial pills.

"*People should be told why it is important to take pills, what the pills are for, and whether the person is sick. This is because you may meet the people while they are not sick and they may wonder why they have to take pills if they are not sick. Then, you should explain what those pills are for. I think that after explanation people will accept to take the pills*" (SSI 08, adolescent, Magude village).

"*First, it would be better to explain what these pills are, their adverse effects: this can happen and that, you can do this at home; advise people that if they feel ill they can go to the hospital, etc. I think the big problem is the adverse reactions of the pills. You should explain to the patient that it may happen, this, this, this. . ., that they shouldn't be alarmed, it's natural, it's the effect of the medication, after a while it may pass, if it doesn't, they can go to hospital*" (SSI 06, healthcare professional, Magude village).

Participants explained that some fieldworkers recommend drunken people to take antimalarial pills, while others do not give pills to drunk people at all. These participants perceived that people should not take pills after drinking alcohol, and they suggested that pills should be left at the household, and people would take in the following day.

"*Usually, the fieldworkers arrive late and find people already drunk. But, some fieldworkers say even if the person is drunk, they recommend him to take the pills. So, we are used to it, that if you have just drunk, you should not take pills. We deny taking the pills after drinking*" (FGD 05, general population, Magude village).

"*If the person is drunk, the fieldworkers should leave the pills, and leave recommendations with a person who is not drunk. He will take it the next day when the drunkenness is finished*" (FGD 06, general population, Magude village).

Participants suspected that not all people comply with the recommended dosage of the antimalarial pills. To overcome this problem, healthcare professionals proposed supervision during the administrations of the pills. They explained that the fieldworkers should visit the households and monitor the compliance of malaria pills intake during the three days of dosage.

"*To comply with the dosage, I think people should take the pills in the presence of the fieldworkers*, *and not let the patient decide to take it in the following days alone. He can have a party and stop taking the pills, and he can take them when he wants. The lack of monitoring can cut the effect of the medicine itself*" (SSI 08, healthcare professional, Magude village health facility).

Several participants appealed to the improvement of fieldworkers' attitudes as they perceived that fieldworkers do not often comply with the local cultural norms such as greeting the members of the households, explaining the reason why they are visiting that household and explaining why and how to take antimalarial pills. Participants expected humble and respectful fieldworkers, and they suggested that fieldworkers should not be young people.

"*Participant 1*: *It is necessary that when a fieldworker arrives at a house he should greet, after he has greeted we will give him chair to sit, and then he communicates to us about the reason why he came to visit us*, *he explains to us how the pills are taken. But there are some fieldworkers who are very young who create difficulties. . .they don't explain, they don't know how to answer adult people.*

*Participant 3: Even if they are not young fieldworkers, some when they arrive they say: "you have to take pills, you also have to take them here", even when someone has asthma, they say: "you have to take, take pills. So, that's what we don't want.*

*Participant 5*: *(. . .) They* [fieldworkers] *should explain their mission well and in a good way so that they can give us pills and we take them*, *in as much as we are satisfied also; they should not prick the heart* [not offen] *the person, because if they prick the heart the person already takes the pills unsatisfied*" (FGD 05, general population, Magude village).

"*Participant 1: Fieldworkers should be people with respect, they should not come with pride, others come with their own problems and put out on me, we will not agree to each other, and some may be sent away.*

*Participant 4*: *A fieldworker has to be someone who works with an open heart and calm*, *so that we can also receive him well*" (FGD 11, general population, Magude village).

Participants also claimed that fieldworkers were outsiders of the community. They proposed training of some local fieldworkers who could understand local language, practices and culture, and who would build a strong relationship with the local communities.

"*Among the fieldworkers, they should include ladies or girls from our area. These people know the local life, it would be simple for them to greet "how are you", have you ever felt something "X"; they would be able to explain the local people in a good manner*" (DGF 07, general population, Mapulanguene).

"*The rfMDA programme should involve local communities; involve someone from the community, it would be better to train someone local that the communities know, it would create confidence in the community, it could be a huge help. The knowledge of that person could help them to join the campaign. Most of the time, it is not because the person does not want to take pills, but the reason is that the fieldworkers distribute the pills and then disappear, they have no connection with the local communities. Some people resist taking pills because of lack of trust to the fieldworkers; because they don't know those people* [fieldworkers]. *The population may think maybe the fieldworkers want to kill them; if someone dies who will they turn to*? *For example, if I am a local fieldworker, I arrive at my neighbour's house, she may even resist*

*a bit to take pills, I try to convince her, (. . .) she ends up having a different idea, and accept. She will think that my neighbour can't give me this to kill me, if she kills me I'll go to her house (. . .). So, if we involve the community a little more, if local people are also into the programme, I think it will be better, we will have a greater adherence, and the programme goals can be achieved*" (SSI 05, healthcare professional, Panjane).

## Discussion

This qualitative study analysed acceptability and perceived barriers to reactive focal mass drug administration (rfMDA) among community members exposed to community engagement campaigns and malaria elimination interventions in rural Magude district. The study found that all group members of the community included in the sample accepted rfMDA regardless the place of residence. This acceptability was associated to the awareness about rfMDA as a result of community engagement campaigns. The perceptions that rfMDA, like the previous MDA, would prevent malaria, improve people's health status, and the fact that the procedures used would reduce the cost incurred by transport to the health facility also influenced rfMDA acceptability. Moreover, participants perceived malaria as a local health concern, and they believed that rfMDA could help to eliminate it. These results are consistent with previous studies in the same study setting [7, 9]. In particular, these previous studies found that high acceptability of MDA was influenced by the perception of malaria as a main health problem [9] and by the community engagement campaign [7]. Moreover, others studies undertaken in Tanzania [17], Eswatini [18] and Cambodia [19] showed that perceived risk for malaria influenced acceptability of malaria treatment.

The results of this study also reveal that the procedures used in rfMDA were accepted despite mixed perceptions about the process of management of pregnancy test outcomes and administration of antimalarial pills to all members of the community. The acceptability of the rfMDA procedures derived from the awareness of the communities that those were recommended procedures to access antimalarial pills; perceptions of the procedures as norms of the health facility, the willingness to know one's health status, and the recognition that malaria could be hidden in the body and transmissible to other members of the community. This result highlights high awareness of malaria transmission and desire of its elimination. Like other studies in the Gambia [20] reported, the acceptance of antimalarial pills without malaria symptoms, may reveal a strong sense of responsibility of the participants of this study toward protecting themselves, their family members and their neighbours.

Despite community acceptability and high awareness of the procedures used in rfMDA, some procedures such as performing malaria tests on children and pregnancy tests were not often welcome, and they could hinder the uptake of rfMDA campaign. The results of this study showed that participants were reluctant to perform malaria tests to children as they perceived it could harm children's health by reducing the amount of blood in their body. In addition, participants were concerned about pregnancy test decision-making and pregnancy testing result disclosure because it could contribute to disagreement among couples, especially when a wife does a test without her husband's consultation, or if other members of the community access the information about a positive pregnancy test before the husband. Moreover, participants had experience of previous antimalarial pills, and they were concerned about drug adverse reactions, and others were reluctant to take drugs without malaria symptoms. These barriers have also been documented in previous studies [18, 21–24]. Furthermore, like previous studies [25] have reported, lack of access to accurate information, spread of misinformation about malaria intervention, being unable to drink alcohol while taking DHAp [7], lack of

trust of fieldworkers, and the demand of food as precondition to take DHAp are potentials barriers to rfMDA.

The barriers identified in this study reflect the need of more community engagement in malaria campaign, which include the community appropriation of the malaria elimination process, involvement of community leaders in the whole process, and training of local community health workers and other local eligible people to serve as fieldworkers. This strategy could contribute to community self-appropriation of the malaria elimination campaign, and it would build a strong relationship between fieldworkers and the community. As the participants suggested, local fieldworkers are more appropriate to work with communities than outsiders as they are more prone to follow and respect the local cultural norms, and this could help to build a strong relationship with the communities.

Community engagement is crucial, and it has been recognised as central to malaria campaign uptake [26, 27]. Several strategies could be used to strengthen rfMDA, including house-to-house visits to inform the population about the planned campaign, and provide non-monetary incentives, such as bed nets, food or school material to children or other things that can incentivise people to participate in the malaria campaign. Incentivising communities has been found as a valid community engagement strategy in a similar campaign in Cambodia [27], where it contributed to the increasing participation of the population in malaria campaign.

## Limitations

This study is limited to the study setting and the selected participants, and the results could not be generalized to other settings. Given to the nature of the qualitative methodology that guided this study, the study sampling was not representative of the study population, and it was subject to sample-bias because only some participants, who were considered as representing specific groups of the community, were selected according to the study objectives. This sample strategy led to exclusion of other community members who could have different views about the study object.

## Conclusion

The community of Magude district found rfMDA and its procedures acceptable as a malaria intervention. This acceptability was associated to rfMDA awareness deriving from community engagement, previous experience of malaria similar campaigns, such as MDA, and willingness of the community to eliminate malaria. However, some barriers, such as lack of decision-making on pregnancy test among women, fear of pregnancy test results, lack of accurate information about rfMDA, fear of DHAp adverse reactions, and reluctance to take drugs without malaria symptoms might affect rfMDA campaign. Thus, there is a need to continue with community engagement and built community self-appropriation of the malaria programme. This could include involvement of local community leaders, before and during rfMDA, and local community health workers and other local people who can work as fieldworkers during rfMDA campaign. Including community's members in rfMDA implementation could optimize rfMDA uptake, and therefore contributing to malaria elimination.

## Supporting information

**S1 Appendix. Semi-structured interview (SSI) guide for household heads, women of reproductive age, adolescents, members of the general community and community leaders (Portuguese version).**
(DOCX)

**S2 Appendix. Semi-structured interview (SSI) guide for household heads, women of reproductive age, adolescents, members of the general community and community leaders (English Version).**
(DOCX)

**S3 Appendix. Semi-structured interview (SSI) guide for healthcare professionals and community health workers (Portuguese version).**
(DOCX)

**S4 Appendix. Semi-structured interview (SSI) guide for healthcare professionals and community health workers (English Version).**
(DOCX)

**S5 Appendix. Focus groups discussion (FGD) guide for general population: men and women (Portuguese version).**
(DOCX)

**S6 Appendix. Focus groups discussion (FGD) guide for general population: men and women (English version).**
(DOCX)

**S1 Table. Consolidated criteria for reporting qualitative studies (COREQ): 32-item checklist.**
(DOCX)

## Acknowledgments

To all study participants in Magude district, we are deeply thankful for accepting to participate in this study and sharing their experiences and views with us. We also address our thanks to the data field team (field supervisors, data collectors and transcribers), and all those that made the reactive focal mass drug administration possible.

## Author Contributions

**Conceptualization:** Beatriz Galatas, Francisco Saúte, Pedro Aide, Khátia Munguambe.

**Data curation:** Carlos Eduardo Cuinhane, Helder Djive, Francisco Saúte, Pedro Aide, Khátia Munguambe, Neusa Torres.

**Formal analysis:** Carlos Eduardo Cuinhane, Helder Djive, Hoticha Nhantumbo, Ilda Murato, Pedro Aide, Khátia Munguambe, Neusa Torres.

**Funding acquisition:** Beatriz Galatas, Francisco Saúte, Khátia Munguambe.

**Investigation:** Beatriz Galatas, Julia Montaña Lopez, Helder Djive, Hoticha Nhantumbo, Ilda Murato, Francisco Saúte, Pedro Aide, Khátia Munguambe.

**Methodology:** Beatriz Galatas, Julia Montaña Lopez, Francisco Saúte, Pedro Aide, Khátia Munguambe.

**Project administration:** Julia Montaña Lopez, Hoticha Nhantumbo.

**Supervision:** Khátia Munguambe.

**Validation:** Khátia Munguambe, Neusa Torres.

**Writing – original draft:** Carlos Eduardo Cuinhane.

**Writing – review & editing:** Neusa Torres.

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
