## [Editor Report · Decision Letter 0]

24 May 2022

PONE-D-22-12696Acceptability and Perceived Barriers to Reactive Focal Mass Drug Administration in the Context of a Malaria Elimination Program in Magude district, Southern Mozambique: A qualitative studyPLOS ONE

Dear Dr. Carlos,

Thank you for submitting your manuscript to PLOS ONE. After careful consideration, we feel that it has merit but does not fully meet PLOS ONE’s publication criteria as it currently stands. Therefore, we invite you to submit a revised version of the manuscript that addresses the points raised during the review process.

We look forward to receiving your revised manuscript.

Kind regards,

Kabiru Abubakar Gulma, Ph.D.

Academic Editor

PLOS ONE

Journal Requirements:

5. "PLOS requires an ORCID iD for the corresponding author in Editorial Manager on papers submitted after December 6th, 2016. Please ensure that you have an ORCID iD and that it is validated in Editorial Manager. To do this, go to ‘Update my Information’ (in the upper left-hand corner of the main menu), and click on the Fetch/Validate link next to the ORCID field. This will take you to the ORCID site and allow you to create a new iD or authenticate a pre-existing iD in Editorial Manager. Please see the following video for instructions on linking an ORCID iD to your Editorial Manager account: " ext-link-type="uri" xlink:type="simple">https://www.youtube.com/watch?v=_xcclfuvtxQ"

6.Please include captions for your Supporting Information files at the end of your manuscript, and update any in-text citations to match accordingly. Please see our Supporting Information guidelines for more information: http://journals.plos.org/plosone/s/supporting-information. 

Additional Editor Comments :

Your paper presents an interesting subject and is, indeed, very sound. However, the punctuation and capitalization errors point to the fact that the paper was not appropriately proofread. As for capitalization, try to employ the use of title case for headings/titles, especially in the reference list. Moreover, your reference list was not strict to the format. I suggest you visit the following link for guides on selecting a particular referencing rule of style and being very strict with its format and punctuation (https://libguides.reading.ac.uk/citing-references/referencingstyles).
---

## [Author Response · Author response to Decision Letter 0]

13 Jun 2022

Subject: Submission of the revised manuscript [PONE-D-22-12696]

Dear academic editor and reviewers,

Thank you for reviewing the manuscript “Acceptability and perceived barriers to reactive focal mass drug administration in the context of a malaria elimination program in Magude district, Southern Mozambique: A qualitative study”. The authors of this manuscript have read the current Instructions for Authors, and agreed to accept the recommedned format. The new manscript version reflects the recommeded format. All authors have also read and agreed upon the submitted version of this manuscript. We believe that the new manuscript will now be suitable for publication format in the PLOS ONE journal.

To acdemic editor:

Answer: We followed the recommeded format and we used the PLOS ONE tampletes to revised the manuscript. The new manuscript reflect the recommened format.

Answer: The research procol was approved by local and national IRB, namely CISM’s institutional ethics committee (CIBS-CISM) and the Mozambican Ministry of Health National Bioethics Committee, and a consent was obtained from parents and guardiens of the minor included in the study. In addition, an assent was obtained from the young adolecents that participated in the study. This information was now added in Methods section, particularly in etihcal consideration section of the new manuscript version. In addition, Table 1 was reformulated for better reading of the presented data.

3. Data availability

We agree and we support the policy of data availability, and we recognize the advantages of data availability. We have read PLOS ONE policy and we think that is very important to share the data publicly. However, the qualitative data used to develop this manuscript involve human discourses, and therefore, there is ethical and legal restrictions to sharing the data publicly. The ethical and legal restriction derive from the fact that the protocol and the informed consent and assent approved by the two ethical review boards referred that the data would only be available to the study team, and the protocol established that all information would be confidential. Thus, no participant of the study was informed that the data would be made publicly. Despite this restriction, the data of this study may be available to all researchers upon request to IRBs. In this regard, we would like to update our statement of data availability to as follows:

Data Availability: The data of this study were collected under individual-level informed consent and assent after a research protocol was reviewed and approved by CISM’s institutional ethics committee (CIBS-CISM) and the Mozambican Ministry of Health National Bioethics Committee. The informed consent signed by the participants stated that: “data will only be available to the study team”, and the protocol stablished that all information will be confidential, and no data from the data collection forms, nor from audio files will be accessible to anyone outside of CISM. Given this statement approved by the two IRBs, data from this study is available upon request to these institutional review boards: CISM’s institutional ethics committee (sozinho.acacio@manhica.net) or the Mozambican Ministry of Health National Bioethics Committee (jflschwalbach@gmail.com) for researchers who meet the criteria for access to confidential data.

Answer: The answer for this question was already provided in question 3.

5. "PLOS requires an ORCID iD for the corresponding author in Editorial Manager on papers submitted after December 6th, 2016. Please ensure that you have an ORCID iD and that it is validated in Editorial Manager.

Answer: The correspondent author already has an ORCID iD, which is 0000-0002-6871-1218. I update this ORCID iD in Editorial Manager page, in Update my information, as recommened.

6.Please include captions for your Supporting Information files at the end of your manuscript, and update any in-text citations to match accordingly. Please see our Supporting Information guidelines for more information: http://journals.plos.org/plosone/s/supporting-information. 

Anwer: The captions were included for the supporting information as recommended. The supported information include the intergiew guides used for data collection (in both Portguese and English) and Table COREQ. It does not include tables or data mentioned in the manuscript.

---

## [Decision Letter · Decision Letter 1]

3 Nov 2022

PONE-D-22-12696R1Acceptability and Perceived Barriers to Reactive Focal Mass Drug Administration in the Context of a Malaria Elimination Program in Magude district, Southern Mozambique: A qualitative studyPLOS ONE

Dear Dr. Cuinhane,

Thank you for submitting your manuscript to PLOS ONE. After careful consideration, we feel that it has merit but does not fully meet PLOS ONE’s publication criteria as it currently stands. Therefore, we invite you to submit a revised version of the manuscript that addresses the points raised during the review process.

 In addition to the minor comments from the reviewers, I realized the authors have used over 30 pages to describe their results. It is important that the authors re-summarize these findings. I have also attached a marked-up revised copy for other comments.

We look forward to receiving your revised manuscript.

Kind regards,

Hammed Mogaji, Ph.D

Academic Editor

PLOS ONE

Journal Requirements:

Reviewers' comments:

Reviewer's Responses to Questions

**Comments to the Author**

1. If the authors have adequately addressed your comments raised in a previous round of review and you feel that this manuscript is now acceptable for publication, you may indicate that here to bypass the “Comments to the Author” section, enter your conflict of interest statement in the “Confidential to Editor” section, and submit your "Accept" recommendation.

Reviewer #1: (No Response)

Reviewer #2: (No Response)

2. Is the manuscript technically sound, and do the data support the conclusions?

Reviewer #1: Yes

Reviewer #2: (No Response)

3. Has the statistical analysis been performed appropriately and rigorously? 

Reviewer #1: Yes

Reviewer #2: (No Response)

4. Have the authors made all data underlying the findings in their manuscript fully available?

Reviewer #1: Yes

Reviewer #2: (No Response)

5. Is the manuscript presented in an intelligible fashion and written in standard English?

Reviewer #1: Yes

Reviewer #2: (No Response)

6. Review Comments to the Author

Reviewer #1: The authors have addressed all questions raised in the last review. However, there are few comments I would love authors to address.

Abstract

1. "The reactive focal mass drug administration (rfMDA) was implemented July 2017 to January 2020". Authors should include in the abstract at what period between July 2017 to January 2020 was study conducted.

Methods

1. In Galatas et al. 2021 in "Community acceptability to antimalarial mass drug administrations in Magude district, Southern Mozambique: A mixed methods study", a population of 48,448 residents, and 10,965 households was provided for a study done during rounds of MDA between November 2015 to February 2017. In this study also conducted in 2017, authors present a population of 63,691inhabitants and 14,583 households.

Can authors give explanation to the difference in population number in studies done within same period?

2." The study was undertaken in September 2017 before the start of the reactive surveillance intervention and continued during the first two months after the start of the intervention".

a. The above contradicts the initial mention that rFMDA started in July 2017. If the people of Magude district did not benefit from intervention until after September 2017, authors should explain this in the Introduction to support what was mention in the Methods section. Otherwise, corrections should be made in the Methods (Study setting) section to agree with the earlier mention.

b. In the Methods section (Study setting), authors should state the duration of study. That is, from September 2017 to when, 2018?

3. "...distributed in 5 Administrative Posts: Magude village, Motaze, Mahele,Panjane and Mapulanguene , and the study covered all these 5 Administrative Posts". In Table 1, some administrative units lacked participants bearing in mind in study purposive sampling was employed. Can authors give reasons why some units lacked particpants?

4. Was the semi-structured interview questionnaire used pre-piloted?

Reviewer #2: (No Response)

7. PLOS authors have the option to publish the peer review history of their article (what does this mean?). If published, this will include your full peer review and any attached files.

Reviewer #1: No

Reviewer #2: No

---

## [Author Response · Author response to Decision Letter 1]

16 Dec 2022

Dear editor and reviewers,

Thank you for reviewing our manuscript “Acceptability and perceived barriers to reative mass focal drug administraction in the context of malaria elimination program in Magude district. Southen Mozambique: a qualitative study”. The authors of this manuscript have read the comments and suggestions, and agreed to accept the recommedned conditions. All authors have also read and agreed upon the submitted version of the manuscript. 

To the Editor:

In addition to the minor comments from the reviewers, I realized the authors have used over 30 pages to describe their results. It is important that the authors re-summarize these findings. I have also attached a marked-up revised copy for other comments.

Answer: The authors re-summarized the findings, and only the essential quotations were left. The new revised manuscript has 28 pages.

Answer: The authors added some data related to financial disclosure, but this did not change the statement at all. We have included the new text of financial disclosure in the cover letter.

Answer to the Reviewer #1:

- Typographic error in the manuscript:

Answer: all typographical errors identified were addressed. Moreover, the authors reviewed the whole manuscripts and corrected all possible grammatical errors.

- Results: Please adapt your tables with examples of PLOS ONE tables. Put full stops too at decimal places and not commas. % is written above and not all over the tables.

Answer: all tables were revised and formatted according to PLOS ONE tables, as presented in the new revised manuscript.

Abstract

1. "The reactive focal mass drug administration (rfMDA) was implemented July 2017 to January 2020". Authors should include in the abstract at what period between July 2017 to January 2020 was study conducted.

Answer: The authors included the period that the study was conducted in the abstract, which was between June and September 2017.

Methods

1. In Galatas et al. 2021 in "Community acceptability to antimalarial mass drug administrations in Magude district, Southern Mozambique: A mixed methods study", a population of 48,448 residents, and 10,965 households was provided for a study done during rounds of MDA between November 2015 to February 2017. In this study also conducted in 2017, authors present a population of 63,691inhabitants and 14,583 households.

Can authors give explanation to the difference in population number in studies done within same period?

Answer: Since the start of the project, the CISM conducted annual census in the district, and Galatas et al. (2021) reported the population of the local census of 2016. However, in the paper, we used the census of the National Institute of Statistic (INE) of 2017 rather than local census because the CISM did not conduct the 2017 local census. We therefore, decided to use the census of INE because we think it is updated compared to the 2016 local census.

2." The study was undertaken in September 2017 before the start of the reactive surveillance intervention and continued during the first two months after the start of the intervention".

a. The above contradicts the initial mention that rFMDA started in July 2017. If the people of Magude district did not benefit from intervention until after September 2017, authors should explain this in the Introduction to support what was mention in the Methods section. Otherwise, corrections should be made in the Methods (Study setting) section to agree with the earlier mention.

Answer: Indeed, there was an error in the text regarding the period of the study and the start of rfMDA intervention. The correct information is that the study was conducted between June and September 2017, while the rfMDA program started in September 2017. We have corrected this information in the revised manuscript.

b. In the Methods section (Study setting), authors should state the duration of study. That is, from September 2017 to when, 2018?

Answer: There was also an error in this section about the study period. The correct information is: the study was conducted between June and September 2017.

3. "...distributed in 5 Administrative Posts: Magude village, Motaze, Mahele,Panjane and Mapulanguene , and the study covered all these 5 Administrative Posts". In Table 1, some administrative units lacked participants bearing in mind in study purposive sampling was employed. Can authors give reasons why some units lacked particpants?

Answer: The study sample size did not cover all participants in all study sites due to unequal distribution of the study participants’ categories and several constraints to accessing the eligible participants. Health professionals and CHWs were not included in some study sites because they were not in all selected communities, and some community leaders were absent during the data collection. Additionally, it was not possible to include adolescents from Mahele and Mapulanguene because the study took place during the school season, and the eligible participants were at schools in different districts. Furthermore, members of the community from Mahele were not included in the sample of semi-structured interviews because they were unavailable due to their agricultural activities. In addition, the lack of accessibility at the selected study sites during the rainy season constrained the eligible participants' access. Moreover, the data collection strategy used also contributed to unequal distribution of the participants in the study setting. The study started by conducting focus group discussions (FGDs) in all study settings and later carrying semi-structured interview. While FGDs were conducted during late months of dry season, the semi-structured interviews were conducted during the start of rainy season, which constrained access to participants in some study settings (Administractive post of Motaze, Mahele and Mapulanguene). we have added an explanation of the sample distribution in the revised manuscript (page 10 of the revised manuscript with track changes).

4. Was the semi-structured interview questionnaire used pre-piloted?

Answer: Yes, all guides were pre-piloted as started in line 205, page 11 of the revised manuscript with track changes.

---

## [Decision Letter · Decision Letter 2]

3 Mar 2023

Acceptability and Perceived Barriers to Reactive Focal Mass Drug Administration in the Context of a Malaria Elimination Program in Magude district, Southern Mozambique: A qualitative study

PONE-D-22-12696R2

Dear Dr.Carlos Eduardo Cuinhane 

We’re pleased to inform you that your manuscript has been judged scientifically suitable for publication and will be formally accepted for publication once it meets all outstanding technical requirements.

Kind regards,

José Luiz Fernandes Vieira

Academic Editor

PLOS ONE

Additional Editor Comments (optional):

Reviewers' comments:

Reviewer's Responses to Questions

**Comments to the Author**

1. If the authors have adequately addressed your comments raised in a previous round of review and you feel that this manuscript is now acceptable for publication, you may indicate that here to bypass the “Comments to the Author” section, enter your conflict of interest statement in the “Confidential to Editor” section, and submit your "Accept" recommendation.

Reviewer #1: All comments have been addressed

Reviewer #2: All comments have been addressed

2. Is the manuscript technically sound, and do the data support the conclusions?

Reviewer #1: Yes

Reviewer #2: Yes

3. Has the statistical analysis been performed appropriately and rigorously? 

Reviewer #1: Yes

Reviewer #2: Yes

4. Have the authors made all data underlying the findings in their manuscript fully available?

Reviewer #1: Yes

Reviewer #2: Yes

5. Is the manuscript presented in an intelligible fashion and written in standard English?

Reviewer #1: Yes

Reviewer #2: Yes

6. Review Comments to the Author

Reviewer #1: All comments raised have been addressed by authors. I recommend the acceptance of the manuscript. Thank you

Reviewer #2: This is an interesting and insightful study. There is great improvement from the first correction. However, few typographical errors need to be corrected.

Additionally, in the results section/discussion, the authors need to be specific and include the percentages (Numbers) and discuss their findings.

References should be revisited and adapted to suit PLOS ONE format.

7. PLOS authors have the option to publish the peer review history of their article (what does this mean?). If published, this will include your full peer review and any attached files.

Reviewer #1: No

Reviewer #2: No

---

## [Editor Report · Acceptance letter]

24 Mar 2023

PONE-D-22-12696R2 

Acceptability and perceived barriers to reactive focal mass drug administration in the context of a malaria elimination program in Magude district, Southern Mozambique: A qualitative study 

Dear Dr. Cuinhane:

I'm pleased to inform you that your manuscript has been deemed suitable for publication in PLOS ONE. Congratulations! Your manuscript is now with our production department. 

Kind regards, 

on behalf of

Dr. José Luiz Fernandes Vieira 

Academic Editor

PLOS ONE